# SWOOSH! RATTLE! THUMP! - ACTIONS THAT SOUND

## ABSTRACT

Truly intelligent agents need to capture the interplay of all their senses to build a rich physical understanding of their world. In robotics, we have seen tremendous progress in using visual and tactile perception; however we have often ignored a key sense: sound. This is primarily due to lack of data that captures the interplay of action and sound. In this work, we perform the first large-scale study of the interactions between sound and robotic action. To do this, we create the largest available sound-action-vision dataset with 15,000 interactions on 60 objects using our robotic platform Tilt-Bot. By tilting objects and allowing them to crash into the walls of a robotic tray, we collect rich four-channel audio information. Using this data, we explore the synergies between sound and action, and present three key insights. First, sound is indicative of fine-grained object class information, e.g., sound can differentiate a metal screwdriver from a metal wrench. Second, sound also contains information about the causal effects of an action, i.e. given the sound produced, we can predict what action was applied on the object. Finally, object representations derived from audio embeddings are indicative of implicit physical properties. We demonstrate that on previously unseen objects, audio embeddings generated through interactions can predict forward models 24% better than passive visual embeddings.

## 1 INTRODUCTION

Imagine the opening of a champagne bottle! Most vivid imaginations not only capture the celebratory visuals but also the distinctive 'pop' sound created by the act. Our world is rich and feeds all of our five senses – vision, touch, smell, sound and taste. Of these, the sense of vision, touch and sound play a critical role in our rich physical understanding of objects and actions. A truly intelligent agent would need to capture the interplay of all the three senses to build a physical understanding of the world. In robotics, where the goal is to perform physical task, vision has always played a central role. Vision is used to infer the geometric shape (Kar et al. (2015)), track objects (Xiang et al. (2015)), infer object categories (Krizhevsky et al. (2012)) and even direct control (Levine et al. (2016a)). In recent years, the sense of touch has also received increasing attention for recognition (Schneider et al. (2009)) and feedback control (Murali et al. (2018)). But what about sound? From the squeak of a door, to the rustle of a dried leaf, sound captures rich object information that is often imperceptible through visual or force data. Microphones (sound sensors) are also inexpensive and robust; yet we haven't seen sound data transform robot learning. There hardly exists any systems, algorithms or datasets that exploit sound as a vehicle to build physical understanding. Why is that? Why does sound appear to be second-class citizen among perceptual faculties?

The key reason lies at the heart of sound generation. Sound generated through an interaction, say a robot striking an object, depends on the impact of the strike, the structure of the object, and even the location of the microphone. This intricate interplay that generates rich data, also makes it difficult to extract information that is useful for robotics. Although recent work has used sound to determine the amount of granular material in a container (Clarke et al. (2018)), we believe there lies much more information in the sound of interactions. But what sort of information can be extracted from this sound?

In this paper, we explore the synergy between sound and action to gain insight into what sound can be used for. To begin this exploration we will first need a large and diverse dataset that contains both sound and action data. However, most existing sound datasets do not contain information about action, while most action datasets do not contain information about sound. To solve this, we create

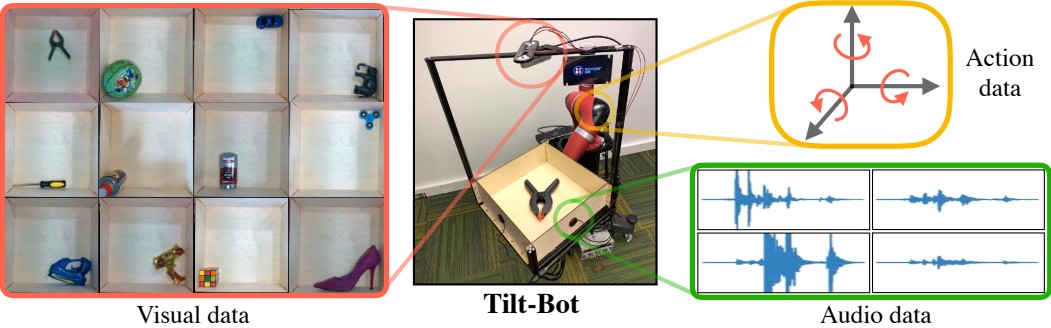

Figure 1: Using Tilt-Bot, we collect 15,000 interactions on 60 different objects by tilting them in a tray. When sufficiently tilted, the object slides across the tray and hits the walls of the tray. This generates sound, which is captured by four contact microphones mounted on each side of the tray. An overhead camera records visual (RGB+Depth) information, while the robotic arm applies the tilting actions through end-effector rotations.

the largest sound-action-vision dataset available with 15,000 interactions on over 60 objects with our Tilt-Bot robot Figure 1. Each object is placed in a tray mounted on a robot arm that is tilted with a random action until the object hits the walls of the tray and make a sound. This setup allows us to robustly collect sound and action data over a diverse set of objects. But how is this data useful? Through Tilt-Bot's data, we present three key insights about the role of sound in action.

The first insight is that sound is indicative of fine-grained object information. This implies that just from the sound an object makes, a learned model can identify the object with 79.2% accuracy from set of diverse 60 objects, which includes 30 YCB objects (Calli et al. (2015)). Our second insight is that sound is indicative of action. This implies that just from hearing the sound of an object, a learned model can predict what action was applied to the object. On a set of 30 previously unseen objects, we achieve a 0.027 MSE error which is 42% better than learning from only visual inputs. Our final insight is that sound is indicative of physical properties of object. This implies that just from hearing the sound an object makes, a learned model can infer the implicit physical properties of the object. To test this implicit physics, we show that a learned audio-conditioned forward model achieves a L1 error of 0.193 on previously unseen objects, which is 24% lesser than forward models trained using visual information. This further indicates that audio embeddings, generated from a previous interaction, can capture information about the physics of an object significantly better than visual embeddings. One could envision using these features to learn policies that first interact to create sound and then use the inferred audio embeddings to perform actions (Zhou et al. (2019)).

In summary, we present three key contributions in this paper: (a) we create the largest sound-action-vision robotics dataset; (b) we demonstrate that we can perform fine grained object recognition using only sound; and (c) we show that sound is indicative of action, both for post-interaction prediction, and pre-interaction forward modeling. Tilt-Bot's sound-action-vision data, along with audio embeddings can be accessed here: https://sites.google.com/view/iclr2020-sound-action.

## 2    THE TILT-BOT SOUND DATASET

To study the relationship between sound and actions, we first need to create a dataset with sound and action. In this section, we describe our data collection setup and other design decisions.

**The Tilt-Bot Setup:** A framework to collect large-scale data needs three key abilities: (a) to precisely control the actions; (b) to be able to interact with a diverse set of objects; (c) to record rich and diverse sound; and (d) requires little to no manual resets. To do this, we present Tilt-Bot (Figure 1). Tilt-Bot is a robotic tray mounted on a Sawyer robot's end-effector. This allows us to precisely control the movements of the tray by applying rotational and translational actions on objects inside it. The tray has dimensions of $30 \times 30$ cm and a payload of 1 Kg allowing us to place a large range of common day objects in it. To collect audio information, four contact microphones (mic) are attached on the four sides of the tray. This allows for the creation of rich audio information from the

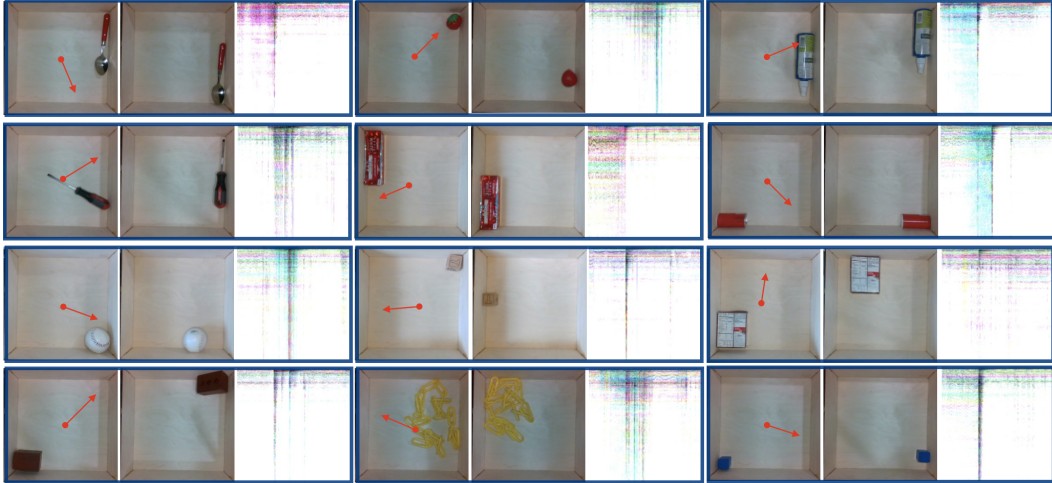

Figure 2: Here are 12 interactions collected using Tilt-Bot. Each interaction is visualized as three images. The left image shows the visual observation of the object before the action is applied along with the applied action. The middle image shows the middle image shows the visual observation after the interaction, while the right image shows the STFT representation of audio generated by 3 of the 4 microphones.

interactions of objects with each other and the tray. To collect visual information, an Intel Realsense Camera (cam) is mounted on the top of the tray to give RGB and Depth information of the object in the tray. Our current setup allows us to collect four-channel audio at 44,100Hz, RGB and Depth at 6Hz, and tray state information (rotation and translation) at 100Hz. Rotational and translational action commands can be sent at 100Hz.

**Data Collection Procedure:** Our dataset consists of sound-action-vision data on 60 objects; 30 of which belong to the YCB object dataset (Calli et al. (2015)), and 30 are common household objects. Details of these objects can be found in Appendix A. For each object, data is collected by first placing it in the center of the tray. Then, Tilt-Bot applies randomly generated rotational actions to the object for 1 hour. We do not apply translational action since we notice minimal motion of the object with it. The rotational actions cause the tray to tilt and make the object slide and hit the walls of the tray. The sound from the four microphones, along with the visual data are continually recorded. Furthermore, using a simple background subtraction technique (Zivkovic (2004)), we can track the location of the object as it collides with the walls of the tray. For every contact made with the tray's wall, which is detected by peaks in the audio stream, we segment a four second interaction centered around this contact. This amounts to around 15000 interactions over 60 robotic hours of data collection. Each of these interactions contain the sound, the RGB+Depth, and the tracking location of the object during the interaction. Examples of the data can be seen in Figure 2. All of our data and pre-processing will be open-sourced, and can be accessed on our website: https://sites.google.com/view/iclr2020-sound-action.

## 3 LEARNING WITH AUDIO

To understand and study the synergies between sound and action, we focus on three broad categories of learning tasks: (a) fine-grained classification (or instance recognition) , (b) inverse-model learning (or action regression) , and (c) forward-model learning . In this section, we will describe our experiments along with insights to better understand the role of sound with action in the context of learning.

### 3.1 PROCESSING AUDIO DATA

Before using audio data for learning, we first need to convert it into a canonical form. Since we will use audio in conjunction with images for several experiments, we build on the representation

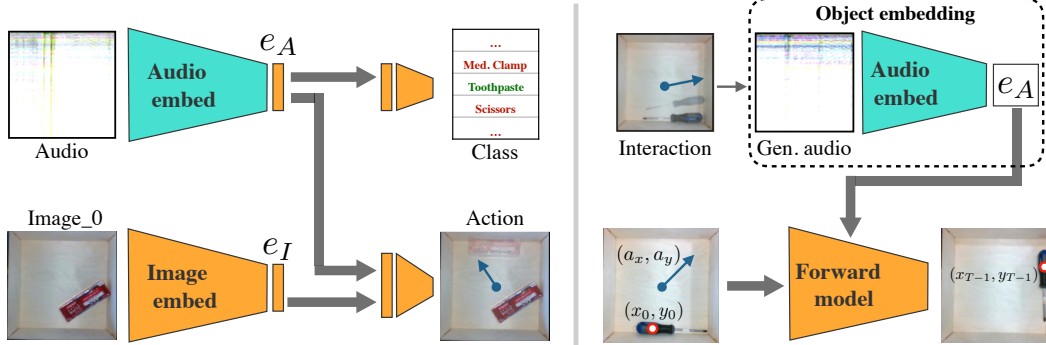

(a) Training the audio embedding

(b) Audio embedding for forward models

Figure 3: To train audio embeddings (a), we perform multi-task learning for instance recognition (top) and action regression (bottom). Once the embedding network is trained, we can use the extracted audio embeddings as object features for downstream tasks like forward model learning (b).

proposed by Zhao et al. (2018). Here the key idea is to convert the high dimensional raw audio (705600 for a 4 second audio recorded at 44.1KHz for 4 audio channels) to a smaller dimensional image. This is first done by subsampling each audio channel from 44.1KHz to 11KHz. Then, a Short-time Fourier transform (STFT) (Daubechies (1990)) with a FFT window size of 510 and hop length of 128 is applied on the subsampled and clipped audio data. For each channel this results in a $64 \times 64$ representation. Stacking the 4 channel audio, we get a $64 \times 64 \times 4$ representation. We further apply a log transformation and clip the representation to between $[-5, 5]$. This representation allows us to treat audio as an image and now effectively run 2D convolutions on audio data, which can capture the temporal correlations from a single audio channel along with the correlations between multiple audio channels. Visualization of this representation can be seen in Figure 2, where the first three channels of audio data ($64 \times 64 \times 3$) are converted to an RGB image.

## 3.2 FINE-GRAINED OBJECT CLASSIFICATION

Classically, the goal of recognition is to identify which object is being perceived. This task is generally done using visual images as input, and is done to test the robustness of visual feature extractors. In our case, we use this task to study what type of object-centric information is contained in sound. For the 60 objects in our TiltBot dataset we first create a training set with 80% of the data and a testing set with the remaining 20%. Then, we train a simple CNN (Krizhevsky et al. (2012)), that only takes the audio information as input and outputs the instance label of the object that generated the sound. This architecture is similar to top part of Figure 3(a).

On our heldout testing set, this trained model achieves a classification accuracy of 76.1%. Note that a random classifier gets a 1.67% accuracy. This shows that audio data contains fine-grained information about objects. Although Owens et al. (2016) demonstrate that audio information can be used to classify broad categories like wood, metal etc., our results show for the first time (to our knowledge) that audio information generated through action gives instance-level information like screwdriver, scissor, tennis ball etc.. To further understand what information sound gives us, we study the top classification errors of our model. In Figure 4 we see that there are two main modes of errors. The first is if instances only differ visually. For example, a green cube cannot be distinguished from a blue cube solely from the sound information. The second error mode is the generated sound is too soft. If the action causes the object to only move a little and not make too much sound, information about the object is masked away and causes classification errors.

## 3.3 INVERSE-MODEL LEARNING

The goal of learning inverse models is to identify what action was applied, given observations before and after the action. From a biological perspective, learning inverse-models implies an understanding of cause and effect, and is often necessary for efficient motor-learning (Wolpert & Kawato

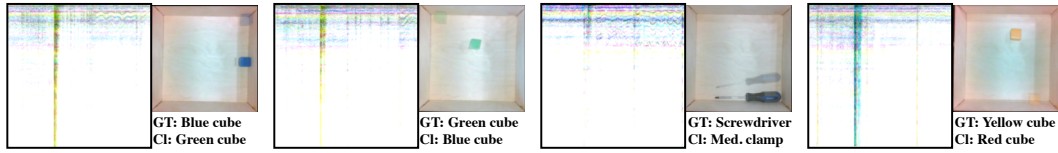

Figure 4: Here we see the top classification errors made by our instance recognition model. There are two key failure modes: (a) misclassifications between similar objects that have different visual properties like the cubes, and (b) when the sound generated is too soft like the misclassification of the screwdriver.

| $\lambda =$ | Train objects | | | | | Test objects | | | | |
| --- | --- | --- | --- | --- | --- | --- | --- | --- | --- | --- |
| | 0.0 | 0.05 | 0.1 | 0.2 | 1.0 | 0.0 | 0.1 | 0.2 | 1.0 | Image |
| Class. (↑) | 0.738 | 0.780 | 0.770 | **0.786** | 0.027 | N/A | N/A | N/A | N/A | N/A |
| Reg. (↓) | 0.395 | 0.024 | 0.022 | 0.014 | **0.008** | 0.352 | 0.027 | **0.020** | 0.027 | 0.043 |

Table 1: Classification and Regression performance across different methods on the Tilt-Bot dataset.

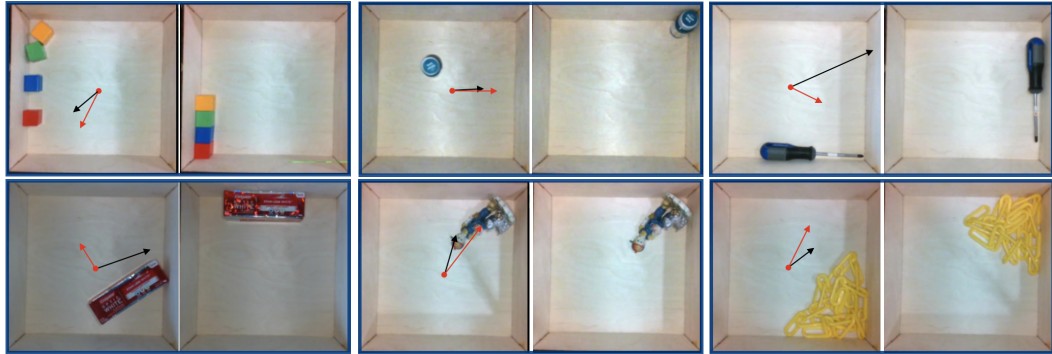

Figure 5: Visualization of predictions from our inverse model. Here, the red arrow represents the ground truth actions taken by the robot, while the black arrow corresponds to the actions predicted by our action-regression model.

(1998)). In the general setting of this problem, a model takes as input the observations before and after an interaction, and outputs the action applied during the interaction. In our case, we want to study if sound contains cause-effect information about actions. Moreover, since inverse-model learning can be evaluated on previously unseen objects, we can test the generalization of audio features not only on objects seen in training, but to novel objects as well.

To demonstrate this, we split our TiltBot objects into two sets: set A and set B, where both sets contain 30 objects with 15 objects from the YCB dataset. Using an architecture similar to the bottom part of Figure 3(a), an inverse model is trained on set A to regress the action. The input into this inverse model is an image of the object before the interaction, and the sound generated during the interaction. Note that the image of the object after the interaction is not given as input. The action that needs to be output is the 2D projection of the rotation vector on the planar tray surface. We evaluate the performance of the inverse model using normalized ($[-1, 1]$) mean squared error (MSE), where lower is better. Testing this model on held-out set A objects, we get a MSE of 0.008, while a random model give a MSE of 0.4. If we use the image of the object after the interaction as input instead of audio, we get a MSE of 0.043. This shows that for these physical interactions using audio information is not just better than random, but in fact better than using visual observations. This insight holds true even when tested on previously unseen set B objects. With set B testing, audio inverse models give a MSE of 0.027, which indicates some amount of overfitting on set A objects. However, this is significantly better than the 0.047 MSE we get from using purely visual inverse models. Sample evaluations of our inverse model can be seen in Figure 5.

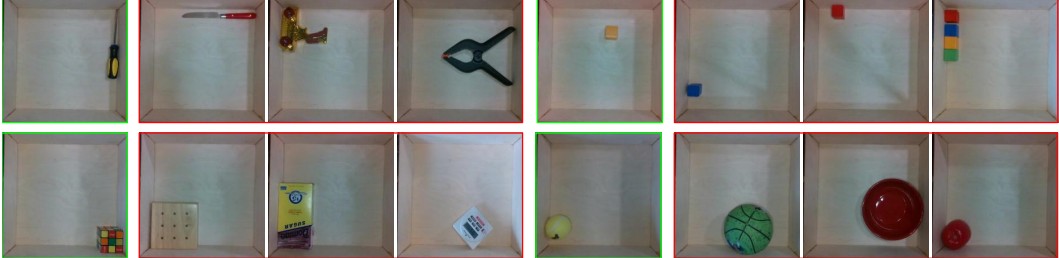

Figure 6: Image retrieval results based on audio embedding. Here, the green box image corresponds to query image and next 3 images in the row (in red) correspond to retrievals.

### 3.4 MULTI-TASK AUDIO EMBEDDING LEARNING

In the previous two sections we have seen how sound contains information about both fine-grained instances of objects and causal effects of action. But what is the right loss function to train an audio embedding that generalizes to multiple downstream tasks? One way would be to train the embedding on the instance recognition task on Tilt-Bot data, while another option would be to train it on the inverse-model task. Both of these tasks encode different forms of information, with classification encoding identifiable properties of the object and inverse model encoding the physical properties of the object. Inspired from work in multi-task learning (Caruana (1997); Pinto & Gupta (2016a)), we take the best of both worlds and train a joint embedding that can simultaneously encode both classification and action information.

As seen in Figure 3(a), the audio embedding $e_A$ is trained jointly using the classification and the inverse model loss according to $\mathcal{L}_{total} = (1 - \lambda)\mathcal{L}_{class} + \lambda \mathcal{L}_{inv}$. Note that when $\lambda = 0$, the embedding captures only classification information, while $\lambda = 1$ captures only inverse-model information. We report the performance of joint learning on held-out data in Table 1. Here, training is performed on set A objects, while testing is done on set A held-out interactions and unseen set B objects. For classification, we find that joint learning improves performance from 73.8% on the 30 set A objects to 78.6%. When trained both set A and set B objects, classification performance improves from 76.1%( Section 3.2) to 79.5%. On inverse-model learning, we notice that joint learning does not improve performance on set A. However, on novel set B objects, we see a significant improvement from 0.027 MSE to 0.020 MSE. Again, this performance is also much better than learning directly from visual inverse-models at 0.043 MSE.

Another way to understand the information captured in our audio embeddings is to look at the top three nearest instance categories given an input object. In Figure 6 we show a few of these object retrievals. Interestingly, these features capture object shapes like matching the long screwdriver to the long butterknife and matching the yellow cube to other colored cubes.

### 3.5 DOWNSTREAM TASK: FORWARD MODEL PREDICTION

Our previous experiments demonstrate the importance of using audio perception. In this section, we investigate if we can use sound to extract physical properties of an object before physically interacting with it. This use case is inspired from recent work on EPI (Zhou et al. (2019)) where probing interactions are used to understand latent factors before implementing the real policy. Here the sound generated through probing interactions would serve as latent parameters for representing the object.

To evaluate the use of audio features for downstream tasks, we perform forward prediction (See Figure 3(b)). Here given an object, a random interaction is performed on it and a sound is generated from this interaction. The embedding network trained using multi-task learning is then used to extract the audio embedding, which will serve as our object's representation. Then, given this object's representation, we can train a forward model that takes as additional input the location of the object and action applied on the object, and outputs the location of the object after the interaction. To learn this forward model, the network has to understand the dynamics of the object. Note that the only object specific information is given through the audio embedding.

|  | Audio Embeddings | | | | | No Audio | |
|---|---|---|---|---|---|---|---|
| $\lambda =$ | 0 | 0.05 | 0.1 | 0.2 | 1.0 | ResNet | Oracle |
| Train Objects | 0.225 | 0.221 | **0.220** | 0.222 | 0.239 | 0.258 | 0.206 |
| Test Objects | 0.195 | 0.194 | **0.193** | 0.1945 | 0.195 | 0.256 | 0.155 |

Table 2: Comparison of using audio embeddings versus pre-trained visual embeddings for forward model prediction. Oracle represents training with object class labels as input.

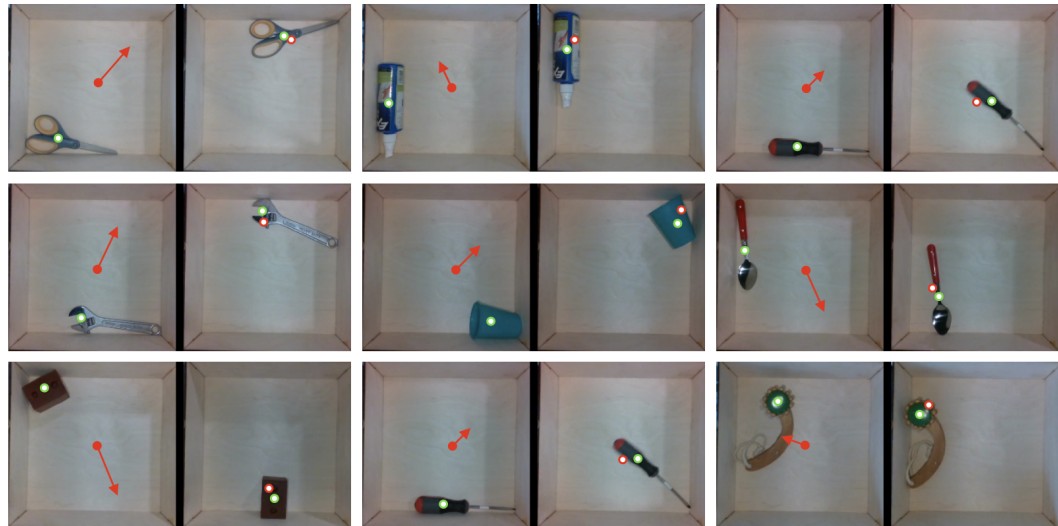

Figure 7: Forward model predictions: Here, green dot on the image represents the ground truth location. Based on the action taken by robot, shown in red arrow in initial state of the tray and corresponding object ground truth location, forward model tries to predict object location which is shown as a red dot.

As seen in Table 2, we report significant improvements in forward model prediction from 0.258 L1 error using visual features to 0.220 L1 error when using the audio embedding feature on objects seen during forward model training. This trend continues for novel set B objects, where both the embedding and forward model was trained on set A objects. Here we see an even larger improvement with visual features giving 0.256 L1 error while audio features giving 0.193 L1 error. This shows that audio embedding information is better able to capture implicit physics of the object as compared to visual features. Moreover, these features are significantly more robust than visual features and also generalize to previously unseen objects.

## 4 RELATED WORK

### 4.1 MULTI-MODAL LEARNING WITH SOUND

Recently, there has been a growing interest to use sound in conjunction with vision, either for generating sound for mute videos, or to localize the part of image that produces sound, or to learn better visual and audio features. Owens et al. (2015) collected hundreds of videos of people hitting, scratching, and prodding objects with a drumstick. This data was then used to train a recurrent neural network which synthesizes sound from silent videos. We collect audio of interactions between objects and a tray. However, instead of relying on humans, which is a huge bottleneck for data collection, we use a robotic platform to collect data. Aytar et al. (2016) use the natural synchronization between vision and sound to learn an acoustic representation using unlabelled two-million videos. Similarly, Arandjelovic & Zisserman (2017a) looks at raw unconstrained videos to learn visual and audio representations that perform on par with state-of-the-art self-supervised approaches. In a similar spirit, we also learn audio representations, albeit through action, to be used for down stream tasks. Arandjelovic & Zisserman (2017b); Senocak et al. (2018) further explores the audio-visual

correspondence in videos to localize the object that sounds in an image, given the audio signal. Zhao et al. (2018) has taken this idea one step further. Given the audio signal, they separate it into a set of components that represents the sound from each pixel. In contrast to these works, we look at obtaining a richer representation of sound by studying its interactions with action.

## 4.2 LEARNING FORWARD MODELS

Standard model-based methods, based on the estimated physical properties and known laws of physics, tries to calculate a sequence of control actions to achieve the goal (Khatib (1987); Murray (2017); Dogar & Srinivasa (2012)). Although this has been widely used for object manipulation tasks in robotics (Cosgun et al. (2011)), manipulating an unknown object is still a daunting task for such methods. This is mainly due to the difficulties in estimating and modeling the novel physical world (Yu et al. (2016)). Given the challenges in predicting the physical properties of novel environment, Deisenroth & Rasmussen (2011); Gal et al.; Amos et al. (2018); Henaff et al. (2017) try to learn dynamic models based on their interactions with the environment. However, when we need to use these learned models on previously unseen objects, these models also fail to generalize. This is because they often do not contain object-specific information. One way to get object-specific information is to use raw visual observations instead of object state (Agrawal et al. (2016); Hafner et al. (2018); Finn & Levine (2017)). In these methods, given the observation of a scene and an action taken by the agent, a visual forward model predicts the future visual observation. These forward models can then be used to plan robotic motions. In our work, we show that instead of using visual information, audio embeddings generated from a previous interaction can be used to improve these forward models.

## 4.3 MULTI-MODAL DATASETS

Alongside algorithmic developments, large scale datesets have enabled the application of machine learning to solve numerous robotic tasks. Several works like Pinto & Gupta (2016b); Levine et al. (2016b); Agrawal et al. (2016); Gandhi et al. (2017) collect large scale visual robotic data for learning manipulation and navigation skills. Apart from visual data, some works Murali et al. (2018); Pinto et al. (2016) have also looked at collecting large-scale tactile data. This tactile or force data can then be used to recover object properties like softness or roughness. Although these datasets contain visual information and action data, they ignore a key sensory modality: sound.

Understanding what information can be obtained from sound requires a large-scale sound dataset. Early attempts (Owens et al. (2015)) collect sound data by recording people interacting with objects. Although this dataset contains large amounts of sound data, it does not contain information about action. In our work, we show that action information not only helps regularize object classification, but also helps in understanding the implicit physics of objects. Prior to our work, Clarke et al. (2018) has shown that sound information is indeed helpful for state-estimation tasks like measuring the amount of granular material in a container. Here, they exploit the mechanical vibrations of granular material and the structure around it for accurate estimation. In our work, instead of a single type of object, we collect audio data across 60 different objects. This allows us to learn generalizable audio features that transfer to previously unseen objects on a variety of tasks like action regression and forward-model learning.

## 5 CONCLUSION

In this work, we perform one of the first studies on the interactions between sound and action. Through our sound-action-vision dataset collected using our Tilt-Bot robot, we present several insights into what information can be extracted from sound. From fine-grained object recognition, to inverse-model learning, we demonstrate that sound can provide valuable information that can be used in downstream motor-control or robotic tasks. In some domains like forward model learning, we show that sound infact provides more information that can be obtained from visual information alone. We hope that the Tilt-Bot dataset along with our findings inspire future work in the sound-action domain, especially in robotic settings where visual data is hard to obtain.

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

## APPENDIX A    DATASET DETAILS

Figure 8 shows image of objects for which the data was collected using Tilt-Bot. Moreover, Table 3 and Table 4 shows the number of interactions collected for each object in set A (seen objects) and set B (novel objects) respectively.

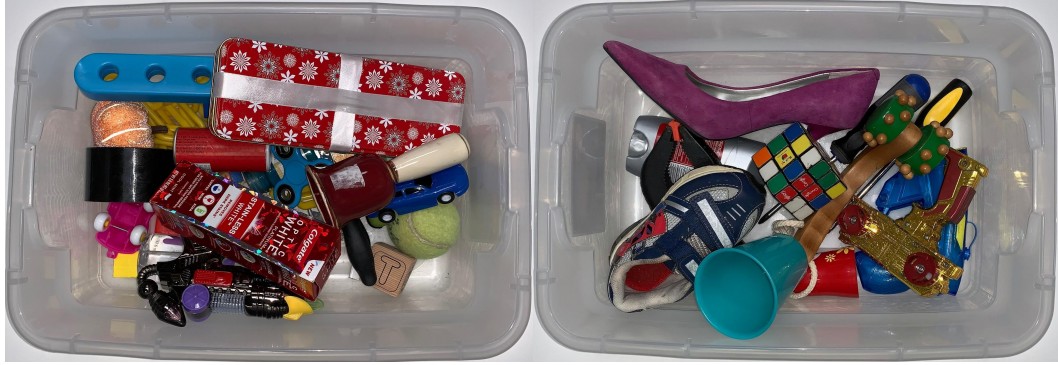

Figure 8: Image of objects for which data was collected using Tilt-Bot.

| Object Name | Number of interactions |
|---|---|
| pudding box | 141 |
| baseball | 365 |
| blue block | 197 |
| salt cylinder | 309 |
| tennis ball | 331 |
| whiteboard spray | 285 |
| toothpaste box | 173 |
| chain | 115 |
| strawberry | 323 |
| flat screwdriver | 324 |
| medium clamp | 172 |
| spoon | 240 |
| rubiks cube | 247 |
| glass | 264 |
| red block | 172 |
| scissors | 216 |
| pear | 336 |
| plum | 331 |
| robot dog with worker | 320 |
| yellow block | 176 |
| phillips screwdriver | 289 |
| acrylic bottle | 365 |
| brown green toy | 171 |
| mug + large clamp + baseball | 612 |
| four cubes | 352 |
| green block | 197 |
| mug | 239 |
| tuna fish can + pudding box + apple | 444 |
| adjustable wrench | 217 |
| green chip clip | 131 |

Table 3: Train Data

| Object Name | Number of interactions |
|---|---|
| timer | 179 |
| tuna fish can | 197 |
| stanley screwdriver | 332 |
| blue fidget | 297 |
| large gift box | 117 |
| rubiks cube jumbled | 131 |
| strawberry | 356 |
| wd 40 | 200 |
| lemon | 389 |
| large clamp | 232 |
| small marker | 307 |
| apple | 229 |
| chocolate mint bar | 105 |
| shiny toy gun | 233 |
| blue nerf gun | 236 |
| blue toy car | 122 |
| green basketball | 414 |
| nine hole peg test | 245 |
| old spice swagger | 199 |
| wine glass | 432 |
| fork | 249 |
| toy elephant | 180 |
| pink heel | 124 |
| rubiks cube | 231 |
| sugar box | 185 |
| bowl | 206 |
| knife | 226 |
| foam brick | 71 |
| soft cube | 67 |
| black tape | 100 |

Table 4: Test Data

