# OpenReview forum: "Swoosh! Rattle! Thump! - Actions that Sound"
_ICLR.cc/2020/Conference — Reject_

### Official Review · AnonReviewer2 · 2019-10-21
**Official Blind Review #2**

**Rating:** 6

**Review:**

This paper explores the interesting connections between action and sound, by building a sound-action-vision dataset with a tilt-bot. This is a good paper overall, I appreciate the efforts on the dataset, and this direction of research is worth pursuing.

Regarding experiments, I like the way it is set up, especially the four microphones, and the action space of the robot. I

A couple of questions:
(1) In the inverse model learning, Fig 3(a) bottom, why are images used as input as well? Don't we want to predict action purely from sound?
(2) In forward model prediction, how are the ground truth locations defined and labeled? Is it the center of mass, and annotated by humans? More details on this experiment will help.


**Experience Assessment:**

I have published in this field for several years.

**Review Assessment: Checking Correctness Of Derivations And Theory:**

I assessed the sensibility of the derivations and theory.

**Review Assessment: Checking Correctness Of Experiments:**

I carefully checked the experiments.

**Review Assessment: Thoroughness In Paper Reading:**

I read the paper at least twice and used my best judgement in assessing the paper.

---

> ### Author Response · Authors · 2019-11-13
> **Response to Reviewer #2**
>
> Thank you for finding our idea of connecting action and sound by using the TiltBot setup interesting, and for appreciating this new research direction.
>
> Why are images used in inverse model: Without the first image, the model would not have enough information to predict the action. This is because different combinations of start-state and action can result in the same generated sound. Using only sound information without start images leads to a lower MSE by around 5-10% across all the inverse model tasks. This shows that using the first image is important, however even without it, we can get fairly high performance. Thank you for suggesting this experiment; we will add this result in the final version of the paper.
>
> Ground truth location of object for forward prediction model: Given an image of an object, we first perform background subtraction to segment the object. The centroid of this segmentation is used as the ground truth state of the object. Additional details will be added in the Appendix and segmentation masks will be released along with the dataset.

---

### Official Review · AnonReviewer1 · 2019-10-23
**Official Blind Review #1**

**Rating:** 3

**Review:**

This paper studies the role of audio in object and action perception, as well as how auditory information can help learning forward and inverse dynamics models. To do this, the authors built a 'tilt-bot', which tilts a box and the object within to collect data (sound & vision) of object interactions. The authors then tested how audio embeddings help object recognition and forward model prediction.

The idea presented in this paper is quite interesting. However, there are no significant technical innovations, the experimental evaluations are quite limited, and the writing can be improved. My overall recommendation is weak reject.

The problem of integrating audio for perception is interesting and has been quite widely explored; however, this paper extends the setup to also explore the effect of audios on dynamics modeling. This is relatively new and may lead to many potential future developments in this direction.

Technically, however, this paper mostly builds on existing technicals on learning forward and inverse models, except that the input is now audio in addition to video. The experimental results are also very limited. They are restricted to a single domain, a fixed collection of objects, and there are no comparisons with published, SOTA algorithms. There are also no results on downstream tasks such as robot manipulation.

I also wonder how the authors think of the related work from Zhang et al: http://sound.csail.mit.edu/ , as they've also studied the effect of auditory and visual data in shape and material recognition.

The writing can be improved. Currently, the model and results are in the same section and mixed together. It'd prefer to separate them. There are a number of typos (incorrect spacing, etc.), especially in Section 3.4. Please double check.


**Experience Assessment:**

I have published one or two papers in this area.

**Review Assessment: Checking Correctness Of Derivations And Theory:**

I assessed the sensibility of the derivations and theory.

**Review Assessment: Checking Correctness Of Experiments:**

I assessed the sensibility of the experiments.

**Review Assessment: Thoroughness In Paper Reading:**

I read the paper at least twice and used my best judgement in assessing the paper.

---

> ### Author Response · Authors · 2019-11-13
> **Response to Reviewer #1**
>
> Thank you for finding our idea interesting and appreciating our new direction!
>
> Novelty/Technical Novelty: Kindly refer to the discussion in global comments on novelty.
>
> Experimental Results: Kindly refer to the discussion in global comments on comparison with SOTA and new experiments on Robotic Manipulation as you asked for.
>
> Writing Style: The choice of having models and results in the same section is a conscious one. Since the goal of this paper is to highlight the synergies between action audio and vision, we believed that a mixed section better reflects our contributions. However, following your suggestions, we are happy to separate section 3 into two separate sections. Thank you for pointing out the typos, this will be fixed in the final version of this work.
>
> Related works: Thank you for pointing out Zhang et al. We think that this a very insightful paper that provides a way to generate audio data for object based on its 3D model and physical properties. However all the data used here is from a simulator. All of our data is from the real-world and on a real robot, because of which our learned embeddings work on real-world tasks.
>
> We hope to have addressed your concerns with additional real-robot experiments. Please let us know if you have any other concerns; we will be happy to answer and clarify.

---

### Official Review · AnonReviewer3 · 2019-10-29
**Official Blind Review #3**

**Rating:** 3

**Review:**

This paper presents audio-visual object classification and motion prediction work on a novel dataset of 60 different objects rolling around in a bin tilted to and fro by a robot, with video and 4-channel audio recordings of the object impacts.   The data is rather novel, is large enough to do ML (around 17 hours of eventful audio/video) and is to be publicly released.  The model architectures  are not of theoretical novelty.   However, the experiments are somewhat interesting.  It was found that the audio contains significant object classification information.  The audio was also good for predicting the trajectory of the object.  This might not be surprising since the microphones are geometrically arranged and may contain directional information along with information about velocity and/or distance traveled.   Overall the experiments are rather thin with only a few experimental results.   A more thorough undertaking might be expected for ICLR papers, with more novel theoretical development and more extensive experiments.

**Experience Assessment:**

I have published in this field for several years.

**Review Assessment: Checking Correctness Of Derivations And Theory:**

I carefully checked the derivations and theory.

**Review Assessment: Checking Correctness Of Experiments:**

I carefully checked the experiments.

**Review Assessment: Thoroughness In Paper Reading:**

I read the paper thoroughly.

---

> ### Author Response · Authors · 2019-11-13
> **Response to Reviewer #3**
>
> Thank you for finding our data novel and our experiments interesting! Indeed, audio contains fine-grained instance level information about objects. It is also not surprising that audio contains trajectory level information since directional information can be inferred from multiple microphones. This is infact how bats echolocate. However, what is really surprising and interesting, is that audio embeddings can be used to infer object properties that can be used for forward physics modeling of the object.
>
> Novelty: Kindly refer to the discussion in global comments on novelty.
>
> Experiments: We have performed two more experiments to drive home the point of the importance of modeling audio, vision and action together. Kindly refer to the global comments for details.
>
> We hope to have addressed your concerns about novelty and experiments with additional real-robot experiments. Please let us know if there are any other specific experiments and comparisons that you were expecting. To our knowledge, this is the first work that combines audio, action, and vision in a single learning framework. If you have any questions, we will be happy to answer and clarify.

---

### Author Response · Authors · 2019-11-13
**Global comments and response to reviewers**

We thank the reviewers for their time and effort. We are also thankful to some interesting suggestions given by reviewers. In global comments, we would like to handle the question of novelty and experiments. For specific questions/queries, we provide individual answers to reviewers.

Novelty: We believe the idea of investigating the relationship between sound and action is novel, which has never been investigated before in this thorough manner. As R1 points out, using sound/audio in forward dynamics is “relatively new and may lead to many potential future developments in this direction.”  Our work creates both a dataset and proposes a new framework for robotic learning with audio, and therefore we believe this work is valuable to the learning community. While some of these findings might be intuitive, our paper is the first to model these and empirically measure the role of audio in dynamics modeling.

Technical Novelty: As a matter of conscious choice, we did not play a lot with forward/inverse models or new architectures to highlight the fact that even while using standard models, using audio in dynamics modeling can lead to significant gains. But the paper still has several technical innovations that might have been overlooked. First, creating a robotic platform and framework to get a dataset with action, audio, and vision is challenging and a novel contribution of this work. Then, using this data to show that audio contains fine-grained information that can distinguish say a screwdriver from a hammer has never been shown before, and the first in this work. Finally, we demonstrate that object representations that we generate solely from audio information is useful for downstream tasks like forward transition model learning and even in robotic manipulation as shown by Robotic Push experiments (See below).

Experiments

R1: Comparison to SOTA
First, we would like to reiterate that this is the first work that combines audio, action, and vision in a single learning framework over a large and diverse variety of objects. Hence, to our knowledge, there are no published work or SOTA algorithms using this new learning framework. If there are any such algorithms and papers, please send them our way, and we will compare with them. In terms of downstream tasks, we perform forward modelling (Section 3.5), which is a precursor for planning in robot manipulation.

R1, R3: More experiments including robot manipulation.

We are pleased to report two new experiments to highlight the strength of our key idea.

Experiment A - Robotic Pushing: In the first set of experiments, we look at how audio embeddings can be used for better robotic pushing. For this, we collect a dataset of around 1000 planar pushing experiments on 10 training set object, and test the audio-conditioned pushing model on 10 testing set objects. We note that without audio embedding the MSE error of the pushing location is 0.180 (normalized coordinates), while using audio embeddings gives an MSE error of 0.159. This clearly demonstrates that audio embeddings can significantly improve robotic pushing. Robot pushing videos can be accessed on our website: https://sites.google.com/view/iclr2020-sound-action.

Experiment B - Few Shot Classification: In the second set of experiments, we look at how audio embeddings can be used for few shot classification on previously unseen objects. The key insight from this experiment is that for all numbers few-shot examples, using audio embeddings gives a performance of around 2-3X the performance of using randomly initialized CNNs. Specifically, say for k=1, using audio embeddings give a performance of 21%, while random CNN gives 7%. This further shows that audio embeddings contains useful information for fine-grained object recognition. Full results can be seen on our website: https://sites.google.com/view/iclr2020-sound-action.

---

### Decision · Program_Chairs · 2019-12-19

**Decision:**

Reject

**Comment:**

This paper investigates using sound to improve classification, motion prediction, and representation learning all from data generated by a real robot.

All the reviewers were intrigued by the work. The paper provides experiments on real robots (never a small task), and a data-set for the community, and a sequence of illustrative experiments. Because the paper combines existing techniques, its main contribution is the empirical demonstrations of the utility of using sound. Overall, it was not quite enough for the reviewers. The main issues were: (1) motion prediction is perhaps expected given the physical setup, (2) lack of comparison with other approaches, (3) lack of diversity in the demonstrations (10 objects, one domain).

The authors added two new experiments with a different setup, further demonstrating their claims. In addition the authors highlighted that the novelty of this task means there are no clear baselines (to which r3 agreed). The new experiments are briefly described in the response (and visuals on a website), but the authors did not update the paper. The new experiments could potentially significantly strength the paper. However, the terse description in the response and the supplied visuals made it difficult for the reviewers to judge their contribution.

Overall, this is certainly a very interesting direction. The results on real world data demonstrate promise, even if they are not the benchmarking style the community is used too.